# Utility of periodic medical questionnaires and examinations for immune-related adverse event screening: A prospective observational study

**Takeshi Azuma**[1]*, **Masato Kano**[1], **Shohei Iwata**[1], **Sachi Honda**[1], **Yuji Miyoshi**[2], **Junko Nishiguchi**[3]

**1** Division of Urology, Tokyo Metropolitan Tama Medical Center, Fuchu, Tokyo, Japan, **2** Department of Rheumatic Diseases, Tokyo Metropolitan Tama Medical Center, Fuchu, Tokyo, Japan, **3** Division of Nursing, Tokyo Metropolitan Tama Medical Center, Fuchu, Tokyo, Japan

* tazuma-tky@umin.ac.jp

**Data Availability Statement:** All relevant data are within the paper and its Supporting information files.

## Abstract

### Background

Immune checkpoint inhibitors (ICIs) are associated with immune-related adverse events (irAEs) specific to the immunity-boosting activity of the drugs and may necessitate discontinuation of treatment depending on their severity. IrAEs may be difficult to diagnose in their early stages as they can occur in any organ. The present, prospective, observational study is the first to attempt to assess the utility of periodic medical questionnaires and laboratory, radiological, and physiological examinations in diagnosing irAEs.

### Methods

We analyzed 51 patients who received immunotherapy for metastatic renal or urothelial carcinoma at Tokyo Metropolitan Tama Medical Center between 2016 and 2020. A medical questionnaire consisting of 41 questions and laboratory tests were administered to the patients on the day of each ICI administration and 1 week afterwards. A significant complaint was defined as a complaint not addressed in the questionnaire immediately prior to the first ICI administration.

### Results

Fifty-one patients with metastatic renal or urothelial carcinoma were enrolled. The mean age was 72.1 years (range: 54–88 years). The male: female ratio was 32: 19. Of the total cohort, 26 (51%) patients had renal carcinoma, and 25 (49%) had urothelial carcinoma. The median follow-up time was 2.6 (range: 0.4–40.7) months. Thirty-three patients (65%) experienced irAEs.

### Conclusions

In our cohort, periodic medical questionnaires and examinations were effective for early diagnosis and prompt treatment of irAEs. Although periodic examinations led to a high irAE

**Funding:** The author(s) received no specific funding for this work.

**Competing interests:** The authors have declared that no competing interests exist.

diagnosis rate, the attendant medical cost was high. Further study is needed to find ways of addressing this issue.

## Introduction

Immune checkpoint inhibitors (ICIs) are effective in treating several types of cancer [1–13]. However, despite their efficacy, they are associated with immune-related adverse events (irAEs), which are specific to the immunity-booting effects of ICIs. Depending on their severity, irAEs may require discontinuation of ICI therapy [14].

IrAEs may have dermatological [15], musculoskeletal [16], endocrinological [17], gastrointestinal [18], renal [19], cardiac [20] or pulmonary [21] manifestations and may be difficult to diagnose in their early stages as they can occur in any organ. Most patients are asymptomatic or have indefinite complaints while others may have carcinoma-like symptoms. Numerous cases of severe irAE and related fatalities have previously been reported. In these reports, the protocol used to evaluate the irAEs was unclear, but the severity of the patients' symptoms prompted exhaustive investigation. Periodic medical questionnaires and examinations can provide a reliable method of mitigating irAEs by enabling prompt diagnosis and uniform treatment. They may also have the added benefit of providing accurate information about irAE incidence.

The present, prospective, observational study is the first to attempt to assess the utility of periodic medical questionnaires and laboratory, radiological, and physiological examinations in diagnosing irAEs.

## Materials and methods

### Patients

The present, prospective, observational study was conducted at Tokyo Metropolitan Tama Medical Center between 2016 and 2020. Fifty-one patients receiving immunotherapy for metastatic renal or urothelial carcinoma were enrolled. This study was approved by the ethical review board of Tokyo Metropolitan Tama Medical Center (30–135) and was conducted in accordance with the principles of the Declaration of Helsinki and the Good Clinical Practice Guidelines. Informed consent was obtained from all the participants.

### Medical questionnaire

Figs 1 and 2 show the medical questionnaire consisting of 41 questions and a reference chart corresponding to each question, respectively. These questions were formulated by referring to reports of irAEs in the Checkmate 025 clinical trial [4]. The questionnaire was administered before the first ICI administration, on the day an ICI was administered, and 1 week later. The pre-ICI responses were used as the baseline against which the responses to the other 2 questionnaires were compared. A significant complaint was defined as a complaint not reported on the first questionnaire. Analysis of the irAEs was facilitated by using a reference chart of irAEs based on the items on the questionnaire (Fig 2).

### Examinations

As with the questionnaire, physiological and laboratory examinations were performed everyday an ICI was administered and 1 week later. Table 1 shows the examination details. A chest Xray and echocardiography was performed monthly and every 3 months, respectively.

**If you are receiving immunotherapy, please answer the following questions.** Date :      /      /

| ① Do you have a dry cough without sputum? | Yes, | from      /      to      / | No |
|---|---|---|---|
| ② Do you feel shortness of breath when walking? | Yes, | from      /      to      / | No |
| ③ Do you feel like you are suffocating? | Yes, | from      /      to      / | No |
| ④ Do you have double vision? | Yes, | from      /      to      / | No |
| ⑤ Is talking difficult? | Yes, | from      /      to      / | No |
| ⑥ Is swallowing difficult? | Yes, | from      /      to      / | No |
| ⑦ Is moving or exerting yourself difficult? | Yes, | from      /      to      / | No |
| ⑧ Do/did you have a headache? | Yes, | from      /      to      / | No |
| ⑨ Has your behavior changed? | Yes, | from      /      to      / | No |
| ⑩ Do/did you have numbness in your limbs? | Yes, | from      /      to      / | No |
| ⑪ Do/did you have motor paralysis? | Yes, | from      /      to      / | No |
| ⑫ Do/did you have sensory paralysis? | Yes, | from      /      to      / | No |
| ⑬ Do/did you have nausea? | Yes, | from      /      to      / | No |
| ⑭ Do/did you have vomiting? | Yes, | from      /      to      / | No |
| If yes, how many times per day?      1 to 2 times   /   3 to 4 times   /   6 times or more | | | |
| ⑮ Do/did you have bloody or black stools? | Yes, | from      /      to      / | No |
| ⑯ Do/did you have diarrhea? | Yes, | from      /      to      / | No |
| If yes, how often?      1 to 3 times   /   4 to 6 times   /   7 times or more | | | |
| ⑰ Do/did you feel physically tired? | Yes, | from      /      to      / | No |
| ⑱ Is your mouth or throat dry? | Yes, | from      /      to      / | No |
| ⑲ Do/did you drink a lot of water? | Yes, | from      /      to      / | No |
| ⑳ Has your urine volume increased? | Yes, | from      /      to      / | No |
| ㉑ Is there any swelling? | Yes, | from      /      to      / | No |
| ㉒ Do you catch colds more easily? | Yes, | from      /      to      / | No |
| ㉓ Do you feel there is something wrong with your awareness? | Yes, | from      /      to      / | No |
| ㉔ Has your urine volume decreased? | Yes, | from      /      to      / | No |
| ㉕ Do/did you find blood in your urine? | Yes, | from      /      to      / | No |
| ㉖ Do/did you sweat a lot? | Yes, | from      /      to      / | No |
| ㉗ Do/did you feel palpitations? | Yes, | from      /      to      / | No |
| ㉘ Do/did you have hand tremors? | Yes, | from      /      to      / | No |
| ㉙ Is your skin itchy? | Yes, | from      /      to      / | No |
| ㉚ Do/did you have a rash? | Yes, | from      /      to      / | No |
| ㉛ Do/did you have blisters? | Yes, | from      /      to      / | No |
| ㉜ Do you have painful mouth ulcers? | Yes, | from      /      to      / | No |
| ㉝ Has your eyesight deteriorated? | Yes, | from      /      to      / | No |
| ㉞ Is your eyesight blurry? | Yes, | from      /      to      / | No |
| ㉟ Do/did you see things that look like flying insects? | Yes, | from      /      to      / | No |
| ㊱ Is everything brighter than normal? | Yes, | from      /      to      / | No |
| ㊲ Do you have dry eyes? | Yes, | from      /      to      / | No |
| ㊳ Do/did your joints hurt? | Yes, | from      /      to      / | No |
| ㊴ Are your joints swollen? | Yes, | from      /      to      / | No |
| ㊵ Are there points or patches of bleeding on your skin? | Yes, | from      /      to      / | No |
| ㊶ Do/did you have bleeding from the gums or inside the mouth? | Yes, | from      /      to      / | No |

**Fig 1. Medical questionnaire.**

**A**

| Question | Interstitial pneumonia | Myasthenia gravis | Encephalitis | Neuropathy | Colitis | Hepatitis | Diabetes mellitus |
|---|---|---|---|---|---|---|---|
| ① Do you have a dry cough without sputum? | ○ | | | | | | |
| ② Do you feel shortness of breath when walking? | ○ | | | | | | |
| ③ Do you feel like you are suffocating? | ○ | ○ | | | | | |
| ④ Do you have double vision? | | ○ | | | | | |
| ⑤ Is talking difficult? | | ○ | | | | | |
| ⑥ Is swallowing difficult? | | ○ | | | | | |
| ⑦ Is moving or exerting yourself difficult? | | ○ | | | | | |
| ⑧ Do/did you have a headache? | | | ○ | | | | |
| ⑨ Has your behavior changed? | | | ○ | | | | |
| ⑩ Do/did you have numbness in your limbs? | | | | ○ | | | |
| ⑪ Do/did you have motor paralysis? | | | | ○ | | | |
| ⑫ Do/did you have sensory paralysis? | | | | ○ | | | |
| ⑬ Do/did you have nausea? | | | ○ | | | | |
| ⑭ Do/did you have vomiting? | | | ○ | | | | |
| ⑮ Do/did you have bloody or black stools? | | | | | ○ | | |
| ⑯ Do/did you have diarrhea? | | | | | ○ | | |
| ⑰ Do/did you feel physically tired? | | | | | | ○ | ○ |
| ⑱ Is your mouth or throat dry? | | | | | | | ○ |
| ⑲ Do/did you drink a lot of water? | | | | | | | ○ |
| ⑳ Has your urine volume increased? | | | | | | | ○ |
| ㉑ Is there any swelling? | | | | | | | |
| ㉒ Do you catch colds more easily? | | | | | | | |
| ㉓ Do you feel there is something wrong with your awareness? | | | | | | | |
| ㉔ Has your urine volume decreased? | | | | | | | |
| ㉕ Do/did you find blood in your urine? | | | | | | | |
| ㉖ Do/did you sweat a lot? | | | | | | | |
| ㉗ Do/did you feel palpitations? | | | | | | | |
| ㉘ Do/did you have hand tremors? | | | | | | | |
| ㉙ Is your skin itchy? | | | | | | | |
| ㉚ Do/did you have a rash? | | | | | | | |
| ㉛ Do/did you have blisters? | | | | | | | |
| ㉜ Do you have painful mouth ulcers? | | | | | | | |
| ㉝ Has your eyesight deteriorated? | | | | | | | |
| ㉞ Is your eyesight blurry? | | | | | | | |
| ㉟ Do/did you see things that look like flying insects? | | | | | | | |
| ㊱ Is everything brighter than normal? | | | | | | | |
| ㊲ Do you have dry eyes? | | | | | | | |
| ㊳ Do/did your joints hurt? | | | | | | | |
| ㊴ Are your joints swollen? | | | | | | | |
| ㊵ Are there points or patches of bleeding on your skin? | | | | | | | |
| ㊶ Do/did you have bleeding from the gums or inside the mouth? | | | | | | | |
| Tests | O₂ saturation | Anti-AchR Abs (Usually negative) | CT | | Diarrhea >4 /day →Consultation for gastrointestinal medicine | AST | BS |
| | Auscultation | CK (Usually high) | MRI | | | ALT | Urinary ketone |
| | KL-6 | CK (Usually high) | | | | Acute exacerbation Re-evaluation after 3-5 days | |
| | Chest X ray | Inspection for myocarditis (High coincidence) | | | | | |
| | CT | | | | | | |

**B**

| Question | Diabetes mellitus | Hypo-thyroidism | Hyper-thyroidism | Hypophysitis | Adrenal insufficiency | Nephritis | Myocarditis | Dermatopathy | Uveitis | Collagen disease | Thrombocytopenic purpura |
|---|---|---|---|---|---|---|---|---|---|---|---|
| ① Do you have a dry cough without sputum? | | | | | | | | | | | |
| ② Do you feel shortness of breath when walking? | | | | | | | ○ | | | | |
| ③ Do you feel like you are suffocating? | | | | | | | ○ | | | | |
| ④ Do you have double vision? | | | | | | | | | | | |
| ⑤ Is talking difficult? | | | | | | | | | | | |
| ⑥ Is swallowing difficult? | | | | | | | | | | | |
| ⑦ Is moving or exerting yourself difficult? | | | | | | | | | | | |
| ⑧ Do/did you have a headache? | | | | | | | | | | | |
| ⑨ Has your behavior changed? | | | | | | | | | | | |
| ⑩ Do/did you have numbness in your limbs? | | | | | | | | | | | |
| ⑪ Do/did you have motor paralysis? | | | | | | | | | | | |
| ⑫ Do/did you have sensory paralysis? | | | | | | | | | | | |
| ⑬ Do/did you have nausea? | | | | ○ | ○ | | | | | | |
| ⑭ Do/did you have vomiting? | | | | ○ | ○ | | | | | | |
| ⑮ Do/did you have bloody or black stools? | | | | | | | | | | | ○ |
| ⑯ Do/did you have diarrhea? | | | | | | | | | | | |
| ⑰ Do/did you feel physically tired? | ○ | ○ | | ○ | ○ | | ○ | | | | |
| ⑱ Is your mouth or throat dry? | ○ | | | | | | | | | ○ | |
| ⑲ Do/did you drink a lot of water? | ○ | | | | | | | | | | |
| ⑳ Do/did your urine volume increase? | ○ | | | | | | | | | | |
| ㉑ Is there any swelling? | | ○ | | | | ○ | ○ | | | | |
| ㉒ Do you catch colds more easily? | | ○ | | | | | | | | | |
| ㉓ Do you feel there is something wrong with your awareness? | | | | ○ | ○ | | | | | | |
| ㉔ Has your urine volume decreased? | | | | | | | ○ | | | | |
| ㉕ Do/did you find blood in your urine? | | | | | | ○ | | | | | ○ |
| ㉖ Do/did you sweat a lot? | | | ○ | | | | | | | | |
| ㉗ Do/did you feel palpitations? | | | ○ | | | | ○ | | | | |
| ㉘ Do/did you have hand tremors? | | | ○ | | | | | | | | |
| ㉙ Is your skin itchy? | | | | | | | | ○ | | | |
| ㉚ Do/did you have a rash? | | | | | | | | ○ | | | |
| ㉛ Do/did you have blisters? | | | | | | | | ○ | | | |
| ㉜ Do you have painful mouth ulcers? | | | | | | | | ○ | | | |
| ㉝ Has your eyesight deteriorated? | | | | | | | | | ○ | | |
| ㉞ Is your eyesight blurry? | | | | | | | | | ○ | | |
| ㉟ Do/did you see things that look like flying insects? | | | | | | | | | ○ | | |
| ㊱ Is everything brighter than normal? | | | | | | | | | ○ | | |
| ㊲ Do you have dry eyes? | | | | | | | | | | ○ | |
| ㊳ Do/did your joints hurt? | | | | | | | | | | ○ | |
| ㊴ Are your joints swollen? | | | | | | | | | | ○ | |
| ㊵ Are there points or patches of bleeding on your skin? | | | | | | | | | | | ○ |
| ㊶ Do/did you have bleeding from the gums or inside the mouth? | | | | | | | | | | | ○ |
| Tests | BS | TSH | TSH | ACTH | Cortisol | Cr | Troponin T >0.1 ng/ml | | | Antinuclear antibody | Plt |
| | Urinary ketone | T3 | T3 | Cortisol | Low Na | Urine test | NTproBNP>2 times higher before treatment→Consultation for cardiovascular medication | | | Rheumatoid factor | PA IgG |
| | | T4 | T4 | Low Na | High K | | | | | | |
| | | | | High K | Low blood pressure | | CKMB | | | | |
| | | | | Low blood pressure | | | Cardiac echo | | | | |

**Fig 2. Reference chart corresponding to each question.**

**Table 1. Examination details.**

| Before immunotherapy |
| --- |
| CBC· Blood picture· AST· ALT· T-Bil· LDH· γGTP· TP· Alb |
| UN· Cre· UA· Na· K· Cl· Ca· CK |
| TSH· FT3· FT4· ACTH· Cortisol |
| Anti-thyroglobulin Abs· Anti-TPO Abs |
| KL-6· SP-D |
| BS· HbA1C |
| ANA· IgG· IgA· IgM· IgE |
| Anti-Ach-R Abs |
| HBs· HBc· HCV |
| PT· APTT· D-dimer NT-proBNP |
| Urine test |
| Two preserved serum samples |
| **During immunotherapy (every month)** |
| CBC· Blood picture· AST· ALT· T-Bil· LDH· γGTP· TP· Alb |
| UN· Cre· UA· Na· K· Cl· Ca· CK |
| TSH· FT3· FT4· ACTH· Cortisol |
| KL-6 NT-proBNP |
| BS |
| Urine test |

## Results

### Patient characteristics

Table 2 summarizes the characteristics of the patients receiving ICI therapy. Fifty-one patients with metastatic renal or urothelial carcinoma were enrolled. Their mean age was 72.1 years (range: 54–88 years). The male-to-female ratio was 32: 19. Of the total, 26 (51%) patients had renal carcinoma, and 25 (49%) had urothelial carcinoma (bladder carcinoma: 13; upper urinary carcinoma: 12). The median follow-up time was 2.6 (range: 0.4–40.7) months.

**Table 2. Patient characteristics.**

| | | Renal Cell Carcinoma | | Urothelial Carcinoma | All Patients |
| --- | --- | --- | --- | --- | --- |
| | | Nivolumab (20) | Nivolumab +Ipilimumab (6) | Pembrolizumab (25) | (51) |
| Age | Average | 72.8 | 72.4 | 71.8 | 72.1 |
| | Range | 56–88 | 56–83 | 54–83 | 54–88 |
| Sex | Male | 11 (55%) | 1 (17%) | 19 (76%) | 31 (61%) |
| | Female | 9 (45%) | 5 (83%) | 6 (24%) | 20 (39%) |
| ECOG PS | 0 | 19 (95%) | 6 (100%) | 25 (100%) | 50 (98%) |
| | 1 | 1 (5%) | 0 (0%) | 0 (0%) | 1 (2%) |
| Clinical Response | CR | 2 (10%) | 0 (0%) | 4 (16%) | 6 (12%) |
| | PR | 2 (10%) | 2 (33%) | 5 (20%) | 9 (18%) |
| | SD | 7 (35%) | 2 (33%) | 4 (16%) | 13 (25%) |
| | PD | 9 (45%) | 2 (33%) | 11 (48%) | 23 (45%) |

ECOG PS: Eastern Cooperative Oncology Group Performance Status.

**Table 3. Immune-related adverse events.**

| Immune-related adverse event | Number of patients |
|---|---|
| Interstitial pneumonitis | 5 |
| Colitis | 6 |
| Hepatitis | 10 |
| Cholangitis | 1 |
| Type 1 diabetes mellitus | 1 |
| Thyroid dysfunction | 2 |
| Isolated ACTH deficiency | 3 |
| Dermatitis | 12 |
| Arthritis | 1 |
| Eosinophilia | 7 |

## Immune-related adverse events

Thirty-three patients (65%) experienced an irAE (Table 3). Table 4 shows the details of the irAEs. Three and eleven patients experienced 3 and 2 irAEs, respectively. In total, 50 irAEs were observed; in 2 patients they were Grade 1, in 20 patients they were Grade 2, and in 1 patient they were Grade 3. Whenever an abnormality was detected via periodic administration

**Table 4. Details of immune-related adverse events.**

| | | Renal Cell Carcinoma | | Urothelial Carcinoma |
|---|---|---|---|---|
| | | Nivolumab | Nivolumab+Ipilimumab | Pembrolizumab |
| **Interstitial Pneumonitis** | | | | |
| | Grade 2 | 3 | | 2 |
| **Colitis** | | | | |
| | Grade 2 | 1 | | 4 |
| | Grade 3 | 1 | | |
| **Hepatitis** | | | | |
| | Grade 1 | 4 | 2 | 2 |
| | Grade 2 | | | 2 |
| **Cholangitis** | | | | |
| | Grade 2 | | | 1 |
| **Type 1 Diabetes Mellitus** | | | | |
| | Grade 2 | | 1 | |
| **Thyroid Dysfunction** | | | | |
| | Grade 2 | | | 1 |
| **Isolated ACTH Deficiency** | | | | |
| | Grade 2 | 2 | | 1 |
| **Dermatitis** | | | | |
| | Grade 1 | 5 | 3 | 5 |
| **Arthritis** | | | | |
| | Grade 2 | | 1 | |
| **Eosinophilia** | | | | |
| | Grade 1 | 1 | 2 | 4 |
| | Grade 2 | 1 | | |
| **Cardiomyopathy** | | | | |
| | Grade 1 | 1 | | |

**Table 5. Association between complaints and immune-related adverse events.**

| Systems | Number of Patients with Complaints | Number of Patients with irAE Diagnosis Based on Complaints |
|---|---|---|
| Respiratory | 2 | 0 |
| Neurological | 8 | 0 |
| Digestive | 7 | 7 |
| Endocrine | 14 | 0 |
| Cutaneous | 12 | 12 |
| Ocular | 3 | 0 |
| Musculoskeletal | 3 | 1 |
| Hematological | 1 | 0 |

of the questionnaire and targeted examinations, more detailed testing was conducted. As a result, almost all irAEs were able to be detected before exceeding Grade 2 severity. There was only 1 case of colitis with Grade 3 diarrhea. Thus, the periodic administration of the medical questionnaire and targeted examinations were useful for detecting irAEs at an early stage and allowing prompt treatment.

## Medical questionnaire

The medical questionnaire was administered 1008 times. Table 5 summarizes complaints following treatment with ICIs. No difference in the distribution of the complaints was found among patients receiving nivolumab, pembrolizumab or ipilimumab. The most frequent complaint was fatigue (14 / 51 or 27%). However, this complaint was not an irAE but a symptom of the cancer. Diarrhea, arthralgia, and itching raised the index of suspicion for colitis, arthritis, and dermatitis, respectively. In 6 patients with diarrhea, all underwent computed tomography and colonoscopy, received the diagnosis of colitis, and were started on oral prednisolone. Arthralgia developed in 1 patient. Although a serological examination returned negative for anti-CCP antibody and rheumatoid factor, joint ultrasonography detected joint effusion, synovial thickness, and power doppler signal (Fig 3). Based on these findings, the patient received a diagnosis of arthritis and started infliximab, methotrexate, and oral prednisolone therapy. Thirteen patients with pruritis received a diagnosis of dermatitis and started steroid ointment therapy. As can be seen, periodic medical questionnaires can play an important role in diagnosing colitis, arthritis, and dermatitis.

## Laboratory examination

Laboratory examinations led to the diagnosis of autoimmune cholangitis, hypothyroidism diabetes mellitus type-1, and isolated adrenocorticotropic hormone (ACTH) deficiency. The patients in whom irAE was diagnosed based on the laboratory examination findings did not experience any of the symptoms associated with the irAEs listed on the medical questionnaire. Prompt follow-up testing on the occurrence of an abnormal finding enabled early diagnosis and timely treatment before symptom development in all the affected individuals.

## Radiological examination

Radiological examination revealed interstitial pneumonitis in 5 patients who did not report any dyspnea-related symptoms on the questionnaire. Oximetry also failed to detect hypoxemia. Based only on the radiological findings, all these patients were successfully treated with oral corticosteroid (1 mg/kg/day), which was tapered as the pulmonary lesions improved.

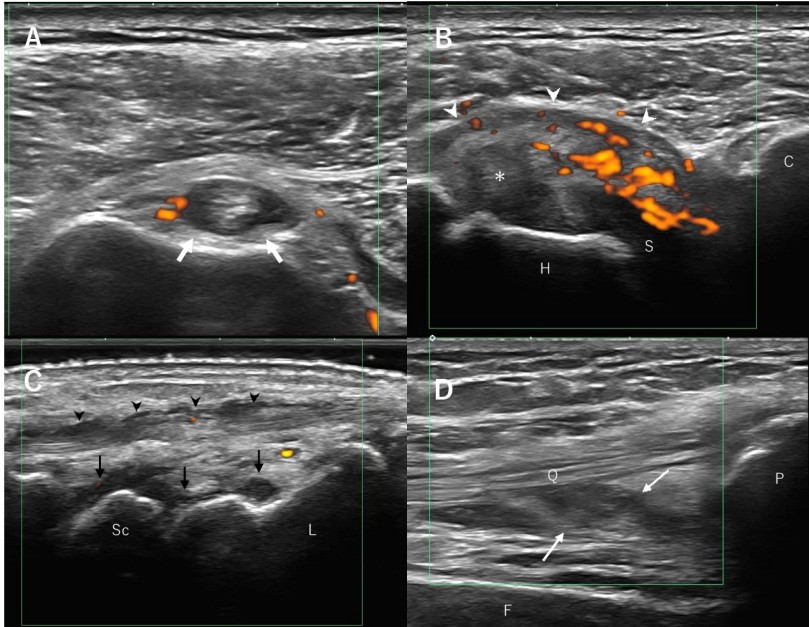

**Fig 3. Musculoskeletal ultrasound images of the patient with polyarthritis induced by immuno-checkpoint inhibitors.** A. Transverse imaging over the bicipital groove of the right humerus shows tenosynovitis of the long head of the biceps brachii (arrows) with moderate effusion and synovial hypertrophy within the tendon sheath. B. Transverse imaging over the right humeral lesser tuberosity shows prominent power Doppler signals extending from beneath the coracoid process over the subscapularis. The subdeltoid bursitis (arrowheads) and the swelling of the long head of the biceps brachii within the rotator interval (asterisk) can be seen. C. Sagittal imaging over the right dorsal wrist shows synovial hypertrophy at the dorsal recesses of the radiocarpal and midcarpal joints (black arrows). Tendinitis surrounding the extensor tendon can be seen (arrows). D. Sagittal imaging of the anterior knee shows hypoechoic to isoechoic synovial hypertrophy within the suprapatellar recess (arrows). H, humeral head; S, subscapularis; C, coracoid process; Sc. Scaphoid; L, lunate; F, femur; P, patella; Q, quadriceps tendon.

## Physiological examination

In 1 patient, periodic echocardiography was able to detect a gradual ejection fraction which decreased by 30% over 3 months. Further investigation revealed cardiomyopathy. The patient was successfully treated with oral corticosteroid (1 mg/kg/day), which was tapered as the symptoms improved.

## Discussion

Periodic screening was able to detect some cases of irAE while the questionnaire and targeted examinations provided clues to detecting other types of irAE. Thanks to the periodic administration of the questionnaire and the examinations, the irAEs were able to be detected at an early stage. Routine laboratory, radiological, and physiological examinations were particularly useful in detecting irAEs in asymptomatic patients. These results indicated that periodic examinations have the potential to detect irAEs before they reach the severe stage. In fact, the actual frequency of irAEs was higher in our cohort than in previous reports.

Periodic administration of questionnaires and examinations offers various advantages. However, each is useful for detecting different types of irAE. The questionnaire was effective in detecting irAEs with characteristics unlike those of sporadic diseases. Only the medical questionnaire was useful for detecting irAEs of this type because these patients never have laboratory examinations findings typical of sporadic diseases. On the other hand, the periodic

examinations were effective in detecting irAEs similar to the symptoms of sporadic diseases, thus enabling their treatment before symptom onset.

In the present cohort, the questionnaire was able to diagnose inflammatory arthritis in 1 female patient. Her serological examination was negative for rheumatoid factor (RF) and anticyclic citrullinated peptide (CCP) antibody, and she had the same titer of antinuclear antibodies (ANA) as before immunotherapy. Cappelli et al. reported 13 patients with inflammatory arthritis caused by immunotherapy [22], all of whom were negative for RF and CCP. ANA was positive in 3 patients, of whom only 1 had a high titer. Changes in ANA before and after immunotherapy were unknown in these 3 patients. Their report suggested that auto-antibodies were not useful for diagnosing inflammatory arthritis. On the other hand, imaging studies, such as ultrasonography or magnetic resonance imaging, are useful for diagnosing arthritis.

Another advantage of the periodic questionnaire was its educational value both for the patients and medical staff. Some of the patients called to report complaints that they had previously seen on the questionnaire, such as diarrhea, joint pain, and skin rash, which can provide some insurance against inexperienced residents or nurses omitting to ask about such complaints in an interview. Educating the patients and the medical staff may thus lead to earlier detection and treatment of irAEs.

Periodic examinations were also very effective in detecting irAEs at an early stage. In our cohort, all irAEs detected in a periodic examination were in their early stages before symptom onset. In the present study, ACTH deficiency was diagnosed in a patient on the basis of the findings of a periodic examination. The diagnosis of isolated ACTH deficiency is usually challenging; as a result, the disease develops insidiously until it causes hypoadrenalism, which in turn can lead to hypoglycemia, hypotension or hyponatremia and become fatal without treatment [23]. The incidence of irAEs associated with the pituitary gland was higher in patients receiving anti-CTLA-4 antibodies than anti-PD-1 antibodies. Several retrospective studies reported a low incidence of pituitary-related irAEs, which had a frequency of 0.5–1.6% and 2.7–5.2% after anti-PD-1 antibody and anti-CTLA-4 antibody therapy, respectively [24,25]. The present, prospective study found the frequency of pituitary-related irAEs to be 6.7% and 0% after monotherapy with anti-PD-1 antibody and anti-CTLA-4 antibody, respectively. The incidence was similar to that reported in a previous, prospective study (9.1%) [21]. Periodic endocrine evaluations were performed as in the previous study although they differed in 2, salient respects, which may have the potential to provide new information. First, endocrine tests were performed more frequently (monthly) in the present study, thereby allowing earlier detection of pituitary-related irAEs. Second, the questionnaire was used to ensure that symptoms, including irAEs, were not overlooked and to determine whether a given symptom was present at the onset of the pituitary-related irAEs.

Periodic examinations have the disadvantage of being costly to perform. Frequent examinations can unnecessarily increase the financial burden on the patients and healthcare system. Although they tend to improve the diagnosis rate by detecting asymptomatic irAEs, the latter may not require treatment. Treatments prescribed on the basis of such findings also may contribute to increasing medical costs unnecessarily.

The present study has a limitation. The study included a widely varied patient population from a single japanese institution and the total number of patients analyzed was relatively small.

## Conclusion

In the present cohort, periodic administration of a questionnaire and target examinations provided various advantages in detecting irAEs. Both were useful for early diagnosis and prompt

treatment. However, the high diagnosis rate inflated medical costs. Further research is necessary to find the optimal balance of diagnosis and treatment-related costs.

## Supporting information

**S1 Data.**
(XLSX)

## Author Contributions

**Conceptualization:** Takeshi Azuma, Shohei Iwata.

**Data curation:** Takeshi Azuma, Masato Kano, Shohei Iwata, Sachi Honda, Yuji Miyoshi, Junko Nishiguchi.

**Formal analysis:** Takeshi Azuma.

**Investigation:** Takeshi Azuma.

**Supervision:** Junko Nishiguchi.

**Writing – original draft:** Takeshi Azuma.

**Writing – review & editing:** Masato Kano, Sachi Honda, Yuji Miyoshi.

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
