## [Decision Letter · Decision Letter 0]

26 Jul 2022

PONE-D-22-18280Utility of periodic medical questionnaires and examinations for immune-related adverse event screening: a prospective observational studyPLOS ONE

Dear Dr. Azuma,

Thank you for submitting your manuscript to PLOS ONE. After careful consideration, we feel that it has merit but does not fully meet PLOS ONE’s publication criteria as it currently stands. Therefore, we invite you to submit a revised version of the manuscript that addresses the points raised during the review process.

We look forward to receiving your revised manuscript.

Kind regards,

Alessandro Rizzo

Academic Editor

PLOS ONE

Journal Requirements:

a) Did participants provide their written or verbal informed consent to participate in this study?

Reviewers' comments:

Reviewer's Responses to Questions

**Comments to the Author**

1. Is the manuscript technically sound, and do the data support the conclusions?

Reviewer #1: Partly

Reviewer #2: Yes

Reviewer #3: Partly

2. Has the statistical analysis been performed appropriately and rigorously? 

Reviewer #1: Yes

Reviewer #2: N/A

Reviewer #3: Yes

3. Have the authors made all data underlying the findings in their manuscript fully available?

Reviewer #1: Yes

Reviewer #2: No

Reviewer #3: Yes

4. Is the manuscript presented in an intelligible fashion and written in standard English?

Reviewer #1: No

Reviewer #2: Yes

Reviewer #3: Yes

5. Review Comments to the Author

Reviewer #1: Despite ICI seem to have finally found their role in genitourinary malignancies, several questions remain unanswered. Among these, the lack of validated biomarkers of response represents an important issue since only a proportion of patients benefit from immunotherapy. Based on these premises, a greater understanding of the role of potential biomarkers including programmed death ligand 1 (PD-L1) expression, tumor mutational burden (TMB), microsatellite instability (MSI) status, gut microbiota and several others is fundamental. In addition, clinical trials on immunotherapy widely differed in terms of drugs, patients, designs, terms of study phases, and inconsistent clinical outcomes. In addition, ICIs have a specific set of toxicities, commonly called immune-related adverse events (irAEs), which are caused by the erroneous activation of the immune system against self-antigens. Multiple organ systems can be affected by irAEs, including the liver, and the incidence of hepatic irAEs varies depending on several factors, including the immunotherapeutic agent, tumor type, and disease setting.

Based on these premises, the study assesses a current, timely topic.

We recommend some changes:

- We believe this article is suitable for publication in the journal although major revisions are needed. The main strengths of this paper are that it addresses an interesting and very timely question and provides a clear answer, with some limitations. Certainly, the study is limited to an Asian population with a very small sample size, and authors should further express this point.

- A linguistic revision is needed.

- Second, the study included a widely varied patient population from a single japanese institution and the total number of patients analyzed was relatively small. Thus, the authors should better highlight the limitations of the current paper.

- The background of the changing scenario of medical treatment in genitoruinary malignnacies should be better discussed, and some recent papers regarding this topic should be included (PMID: 33714725; PMID: 34894318).

Major changes are necessary.

Reviewer #2: Major Correction:

1. The Statement required evidence: Please provide imaging results.

“Changes in ANA before and after immunotherapy were unknown in these three patients. Their report suggested that auto-antibodies were not useful for diagnosing inflammatory arthritis. On the other hand, imaging studies, such as ultrasonography or magnetic resonance imaging, are useful for diagnosing arthritis.”

Minor Corrections:

Several minor corrections have been detected and highlighted within the main manuscript PDF file. Please go through the Spacing related corrections.

Recommendation: Please use a tool that can potentially detect and help to correct grammatical errors.

Reviewer #3: This is a fairly well written manuscript. As the authors have mentioned in the drawbacks, the practical adaptation and widespread implementation of questionnaires and examinations may not really be feasible. However, the authors present interesting correlations that can be useful for diagnosis and treatment making this manuscript and the overall study useful for the scientific community.

- It's not exactly clear from the manuscript the usage of the term "significant complaint". While it is described a few times across the manuscript, it is not clear how it is used in interpretation of the results or the discussion. Please clarify.

- Please change all numbers that are spelt out into numerical. Only a number at the beginning of the sentence needs to be spelt out. Others need to be numerical.

6. PLOS authors have the option to publish the peer review history of their article (what does this mean?). If published, this will include your full peer review and any attached files.

Reviewer #1: No

Reviewer #2: **Yes: **Shovan Dutta

Reviewer #3: No

---

## [Author Response · Author response to Decision Letter 0]

26 Aug 2022

Major Correction:

1. The statement required evidence: Please provide imaging results.

“Changes in ANA before and after immunotherapy were unknown in these three patients. Their　report suggested that auto-antibodies were not useful for diagnosing inflammatory arthritis. On　the other hand, imaging studies, such as ultrasonography or magnetic resonance imaging, are　useful for diagnosing arthritis.”

Thank you for your remarks. We added an ultrasonography image in Figure 1.

Minor Corrections:

Several minor corrections have been detected and highlighted within the main manuscript PDF

I could not find this PDF. We asked our in-house, native English-speaking editor to edit the manuscript. The editor was not mentioned in the Acknowledgements section because this requirement is not applicable.

---

## [Editor Report · Decision Letter 1]

30 Aug 2022

Utility of periodic medical questionnaires and examinations for immune-related adverse event screening: a prospective observational study

PONE-D-22-18280R1

Dear Dr. Azuma,

We’re pleased to inform you that your manuscript has been judged scientifically suitable for publication and will be formally accepted for publication once it meets all outstanding technical requirements.

Kind regards,

Alessandro Rizzo

Academic Editor

PLOS ONE

---

## [Editor Report · Acceptance letter]

19 Sep 2022

PONE-D-22-18280R1 

Utility of periodic medical questionnaires and examinations for immune-related adverse event screening: a prospective observational study 

Dear Dr. Azuma:

I'm pleased to inform you that your manuscript has been deemed suitable for publication in PLOS ONE. Congratulations! Your manuscript is now with our production department. 

Kind regards, 

on behalf of

Dr. Alessandro Rizzo 

Academic Editor

PLOS ONE